# Effect of Fire Temperature and Exposure Time on High-Strength Steel Bolts Microstructure and Residual Mechanical Properties

**DOI:** 10.3390/ma14113116

**Published:** 2021-06-06

**Authors:** Paweł Artur Król, Marcin Wachowski

**Affiliations:** 1Division of Concrete and Metal Structures, Faculty of Civil Engineering, Institute of Building Engineering, Warsaw University of Technology, 16 Armii Ludowej Ave., 00-637 Warsaw, Poland; 2Faculty of Mechanical Engineering, Military University of Technology, 2 Gen. S. Kaliskiego St., 00-908 Warsaw, Poland; marcin.wachowski@wat.edu.pl

**Keywords:** effect of temperature, exposure time, steel microstructure, residual mechanical properties, high-strength steel bolts, heat treatment of steel, phase transformation, fire, cooling method

## Abstract

In this study, the influence of different fire conditions on tempered 32CrB3 steel bolts of Grade 8.8 was investigated. In this research different temperatures, heating time, and cooling methods were correlated with the microstructure, hardness, and residual strength of the bolts. Chosen parameters of heat treatments correspond to simulated natural fire conditions that may occur in public facilities. Heat treated and unheated samples cut out from a series of tested bolts were subjected to microstructural tests using light microscopy (LM), scanning electron microscopy (SEM), energy dispersive spectroscopy (EDS), XRD phase analysis, and the quantitative analysis of the microstructure. The results of the microstructure tests were compared with the results of strength tests, including hardness and the ultimate residual tensile strength of the material (UTS) in the initial state and after the heat treatments. Results of the investigations revealed considerable microstructural changes in the bolt material as a result of exposing it to different fire conditions and cooling methods. A conducted comparative analysis also showed a significant effect of all such factors as the temperature level of the simulated fire, its duration, and the fire-fighting method on the mechanical properties of the bolts.

## 1. Introduction

The behaviour of steel structures during and after fire has stimulated the imagination of researchers and engineers for decades. Reflections derived from the observation of real fires in steel-structure buildings lead to the conclusion that disasters and collapses of these buildings most often happen not during the fire flashover, but during the fire extinction phase. When structural elements previously subjected to sufficiently long fire exposure lose or shorten their stability, this generates additional tensile forces that were not previously present. In such circumstances, internal forces are redistributed or connections between structural components are overloaded. At the same time, provided that the fire temperature has reached a sufficiently high level, microstructural changes can be seen in the structural materials. Observation of these structural destruction mechanisms permitted an assumption that high-strength bolts, which were subjected to heat treatment at the production process stage, lose their strength and load-bearing properties much faster than members made of conventional steel, thus determining the safety of steel constructions.

Brian Kirby is one of precursors of modern research on post-fire behaviour and residual mechanical properties of bolts. In [1,2], Kirby presented results of research carried out on M20-8.8 bolts made during various production processes—using hot and cold forging. He subjected the bolts to pre-heating in the temperature range of 20–800 °C, maintained in a given temperature level for 60 min, and then naturally cooled them in air. He observed that hot-forged bolts are more sensitive to temperature changes than cold-forged ones. As a result of his research, he proposed using crystallographic methods to develop a method enabling identification of the maximum temperature that bolts could reach during a fire. He observed that post-fire residual strength of steel can be determined, inter alia, by specifying its hardness on a polished surface using the Vickers hardness tests with a load of 294.2 N (HV30) and comparing results obtained with those of a static tensile test, further bearing in mind that there is some orderly relationship between the hardness of steel and its tensile strength. He pointed out that if bolt production process parameters are known, this knowledge can be used to assess the temperature reached by a bolt during a fire by comparing its residual hardness with hardness of the material of the bolt in its initial state. At the same time, he demonstrated that if the fire temperature level exceeds the tempering temperature in the production process, the bolt material will soften. He showed that knowledge of metallurgical changes occurring in the bolt material can be helpful in diagnosing fire-damaged buildings.

The behaviour of bolts during a fire was also analysed, among others, by Gonzalez et al. [3] who focused their research on tensile tests of grade 10.9 bolts. Tests limited to destructive testing (static tensile test and static shear test) reflecting the behaviour of bolts under natural fire conditions (comprising the cooling phase) were also carried out by Hanus et al. [4]. They confirmed that if the structure is not destroyed during the temperature elevation phase, tensile forces generated in axially-restrained beams during the cooling phase may lead to the failure of bolted joints. Unfortunately, they did not conduct any research related to the analysis of material structure changes.

A number of works by Kodur et al. [5,6] are also worth paying greater attention to. In [5], the authors analyse the effect of temperature on variability of thermal and mechanical properties of high-strength steel bolts. They observed bolt failure mechanisms occurring during the tensile test through the prism of bolt microstructure that affected the material ductility and the failure model, as well as the shape of the fracture surface after failure. In [6], apart from routine destructive testing aimed at the assessment of residual mechanical properties of grade 8.8 bolts subjected to heating and controlled cooling cycles, they further devoted more attention to the issues of crack propagation and bolt failure models, conducting a broader analysis of fracture surface shapes obtained in the context of the target heating temperature level. They made an attempt to explain how cracks propagate depending on the microstructure of the bolt material. In [7], Yahyai et al. undertook similar research as in [6], focusing on bolts with a higher strength grade—10.9. The effects of a heating temperature, the chemical composition of the charge steel (raw material), and production process parameters on durable mechanical properties were subject to an in-depth analysis. A considerable part of the work was devoted to a detailed analysis on the surface of bolt failure, explaining the shape and form of fractures, and changes occurring in the steel microstructure.

It is worth mentioning that the available sources of knowledge present some research results relating only to two strength classes of bolts—grade 8.8 [1,2,4] and grade 10.9 [3,7]. The results obtained for bolts of grade 10.9 are qualitatively and quantitatively comparable to those presented for bolts of grade 8.8. Due to the fact that the bolts of strength classes lower than 8.8 are currently used only for connecting secondary elements, which are usually permanently deformed during a fire, it seems unreasonable to conduct fire tests using these bolts. It is assumed that such bolts will be replaced after a fire, including any damaged part to which they have been attached. In addition, due to the relatively low risk of a fire in a building during its technical life, so far—mostly for economic reasons—the widespread use of stainless steel bolts or special alloy steel bolts, including fire-resistant and creep-resistant ones, has not been adopted. These materials have a slightly different electrostatic potential, which would also require additional measures to prepare the contact surfaces of these bolts with structural steel elements in order to prevent accelerated galvanic corrosion. In current literature, however, one can find research devoted to fire tests of this type of bolts [8].

Some more interesting works from the borderline of material engineering and fire safety engineering, which are related to the issues discussed in this article, include the paper by Chi and Peng [9]; the authors present the results of steel plate tests that were heated to a temperature of 800 °C and higher and then quickly cooled in water. The work was aimed at demonstrating the possibility of reconstructing a fire scenario in a post-fire investigation on the basis of changes in the material microstructure in a situation where fire development and the level of the temperature reached in a fire were not known. The plates underwent, inter alia, metallographic tests to analyse their composition and microstructure and the results obtained were compared to the mechanical parameters of tested members destroyed by fire. The research included a detailed quantitative analysis of individual phase components. As a result of the heat treatment, the pearlite phase disappeared completely, ferrite was reduced from 80% to 30%, bainite increased its share to 30%, and martensite increased to 40%. A significant increase observed in the martensite phase changed the structure of the steel plate damaged by fire, which was reflected in the change of its material properties. Although yield point and tensile strength showed a growing trend, ductility of the member dropped significantly from 32.5% to 15%, which may result in a greater likelihood of sudden steel failure in structural elements of a building and translate into a decreased safety of its use.

Analyses of the effect of microstructure on mechanical properties of post-fire structural steels were also undertaken by Sajid et al. [10]. As part of the research, they analysed samples made from three grades of structural steel, subjected to heating in the temperature range from 500 to 1000 °C for 60 min, and then naturally cooled in the air. They presented changes in the microstructure in relation to the heating temperature. The analyses they conducted led to a conclusion that an increase in the share of ferrite fraction and the ferrite grain size leads to a decrease in the residual post-fire yield point and tensile strength, as well as an increase in the ductility of the tested structural steel grades. Based on the obtained results, the authors proposed multivariate linear regression equations to estimate post-fire/residual yield point of steel as a function of ferrite grain size and pearlite colony size. In their opinion, the results of these tests may be useful for future engineers and can be used to assess the quality and strength parameters of post-fire steel based solely on the results of microstructural tests, especially in situations where information of the temperature level reached in a fire is not available.

Works by Haiko et al. [11] and Xie et al. [12] are also worth mentioning. However, since they do not relate directly to the issues addressed in this article, they will not be broadly discussed.

The literature review shows that there is still a gap in the field of post-fire tests of steel structures devoted, in particular, to the selected components of connections and joints. In current published research, no attempt was made to determine the effect of the time duration of a developed fire or the applied fire-fighting strategy on the mechanical properties of the connectors, especially in close correlation with their microstructural changes. The aim of this paper is to fill this gap at least partially.

This article presents the results of research conducted to investigate the effect of various thermal and environmental conditions, typical for a real fire situation, on changes in the mechanical properties of high-strength construction bolts that results from their microstructural transformation. M20-8.8 bolts were used for the research for they are commonly used in prestressed butt joints and friction lap joints and even more commonly—due to their universal properties—used in regular non-prestressed joints. During the preliminary phase of testing, the bolts were subjected to thermal effects corresponding to selected conditions of a simulated fire, by heating them in batches in an electric furnace for the time specified in [13]. Following the exposure to high temperature, some bolts were set aside to cool down naturally. The intention was to recreate conditions of a spontaneous and natural fire, resulting either from a shortage of combustible substances or an insufficient amount of oxygen. The other part of the bolts was shock-cooled by immersion in water, which corresponded to a simulated firefighting operation carried out by rescue and firefighting teams. The time of heating corresponded to fire safety requirements adopted within the EU and established by law in relation to structural elements of buildings and building structures. Wording of national legal acts [13,14] directly implements the provisions of Regulation (EU) of the European Parliament and of the Council in 9 March 2011 [15], laying down harmonised conditions for the marketing of construction products. Their purpose is to ensure that basic requirements, such as required load-bearing capacity and structural stability, fire safety, health, and safety of use, are met.

## 2. Materials and Methods

Samples for microstructural tests were cut from a series of bolts previously subjected to fire exposure in various thermal conditions and, alternatively, also a simulated firefighting operation. The bolts were subjected to heating/tempering processes at various temperatures (400, 600, 800, and 1000 °C). In the case of the first series of samples, cooling was carried out naturally in the air (air-cooling/sample symbol: AC) by allowing the samples to cool down slowly, while in the case of the second series cooling was performed with an accelerated method which is immersion in water (water-cooling/sample symbol: WC). In each of the cases, two different heating times of 60 min and 240 min, respectively, were applied corresponding to the selected requirements resulting from [13]. A list of samples is presented in Table 1. Apart from the reference sample in the initial state (IS), the samples were labelled according to the X/Y/Z principle, where X denotes the cooling method, Y denotes the heating temperature, and Z denotes the heating time at a given temperature. Microstructural tests and hardness measurements were carried out using samples cut perpendicularly to the bolt shank axis, the area of which corresponded to the shank cross-section in the threadless place.

The bolts were made of an alloyed steel with an addition of boron, designated as 32CrB3, for which the chemical composition is provided according to the manufacturer’s quality certificate in Table 2.

In the production process, the bolts were made of smooth wire rod in the hot-forging process and then subjected to thermal improvement by quenching at a temperature of approximately 850–860 °C and tempering at a temperature of approximately 550 °C, which resulted in the obtaining of expected mechanical properties that corresponded to grade 8.8. Cut out samples for microstructural testing were hot-mounted in phenolic resin with Struers Multifast graphite filler and then ground using grinding wheels of 320, 600, 800, and 1200 gradation. The samples prepared in this manner were polished using a diamond suspension with a grain diameter of 3 μm and 1 μm. The microstructure of the steel was revealed by etching with a 2% solution of nitric acid in ethanol (2% NITAL).

In order to illustrate the microstructural changes resulting, inter alia, from phase transformations, the research material was analysed using the OLYMPUS LEXT OLS4100 digital light microscope and the JOEL scanning electron microscope (SEM), model JSM-6610, equipped with a secondary electron detector (SE) and a backscattered electron detector. The light microscopy tests were performed on the sample series shown in Table 1.

At the stage of microstructural tests, samples heated at the temperature of 400 °C were eliminated since the results of the static tensile test and the hardness test did not reveal any noticeable changes in relation to the initial state IS. This can be explained by the fact that this temperature is significantly lower than the tempering temperature used during the production process and the phase transformation temperature of steel, A_1_. In the case of each of the samples, photos were taken at several points along the shank width, corresponding to their distance from the cross-sectional edge representing 1, 3, 5, 7, and 9 mm, respectively. For each of the points, two photos were taken at ×50 and ×100 magnifications, respectively, to obtain a more complete picture of changes occurring in the material microstructure. A scanning electron microscope (SEM) was used to test the initial state sample, with an accelerating voltage of 20 kV. The microscopic observations were supplemented with surface microanalysis of the chemical composition by means of the Oxford X-Max energy dispersive X-ray spectrometer (EDS), the results of which were generated in the form of an X-ray spectrum. In order to demonstrate the presence of carbides in the material structure, phase analysis was performed with the use of X-ray diffraction (XRD).

Images of the bolt microstructure in the form of photos obtained with the use of a digital light microscope at the magnification of ×100 were used for a quantitative analysis. Photos taken at a distance of 3 mm and 7 mm from the sample edges were selected for this purpose in order to capture differences, if any, in the microstructure along the width of the bolt shank. These images were used to outline grains of two basic phases, namely ferrite and pearlite, observed in the material microstructure. The quantitative analysis was performed using the MountainMap program. For each of the phases parameters such as the number of grains visible in the photo, their density, grain surface, or the mean equivalent grain diameter were determined. The types of samples utilized in this type of analysis are presented in Table 1. Considering the size of this article, results of the quantitative analysis were not presented in detail and only some of them are shown in Table 10 and Figure 1, which are included further in the work.

The Vickers HV hardness tests were carried out with a load of 294.2 N in the same places along the shank width and compared with the residual tensile strength values obtained from the previously performed static tensile test in order to confirm the correlation relationship between hardness HV and ultimate tensile strength UTS. For structural steels, it is presented in various literature sources in the form of the following relationship.
UTS [MPa] ≈ (3.2 ÷ 3.5)·HV(1)

In order to eliminate human error, the hardness tests were carried out in an automated manner using the NEXUS 4300 stationary hardness tester in several places along the width of the bolt shank in order to capture any differences. Due to the lack of differentiation of the measurement results across the width of the sample, the mean value was taken as representative for further considerations.

## 3. Results

### 3.1. Microstructural Analysis

The analysis was aimed at investigating and assessing the effect of the secondary heat treatment resulting from exposure to various thermal conditions that may occur during a fire and the accompanying firefighting operation on the microstructure and mechanical properties of the bolt steel, which was previously quenched and tempered during production processes. The material in the initial state IS is characterised by a martensite structure (α phase—dark phase/areas in the photos of the IS sample) with a small quantity of residual austenite (γ phase—light areas in the photos of the IS sample) (Table 3).

Analysis of the photos included in Table 3 does not show any significant differences between the microstructure image seen in the photo taken at a distance of 3 and 7 mm, respectively, from the shank edge. The presence of the martensite structure is related to the quenching process previously carried out, which is typical of steel bolt products and particularly those with increased strength. The martensite structure occurs throughout the cross-section of the tested element along the entire width of the bolt shank.

Due to the 0.3% carbon content, the steel covered by the research is referred to in the literature as hypoeutectoid steel and, in its unquenched state, is characterised by a ferrite and pearlite structure. Heat treatment of steel in the quenched state is called tempering and consists in heating up of steel to a temperature below the critical value A_1_ = 727 °C (read out from the Fe-C phase diagram), leaving it at this temperature, and slowly cooling to ambient temperature. Generally, depending on the heating and cooling rate as well as the amount and type of alloying elements in low-alloy carbon steels, the critical temperature may somewhat vary [17]. Heat treatment below temperature A_1_ does not lead to the formation of austenite (γ phase), while annealing above A_1_ takes place when the ferrite (α phase) and austenite (γ phase) states coexist. Processes that occur during tempering are closely related to the phenomenon of diffusion of carbon and alloying elements, therefore they depend on both the temperature level and the heat treatment time. The purpose of the annealing process is to obtain a more fine-grained and more plastic structure, which is desirable in the context of more predictable behaviour and a non-brittle model of structural failure. Fine graining leads to an increase in the yield point since a denser mesh of grain boundaries is shaped in the structure of the material, which may slow down the formation of dislocations in the steel crystal lattice. According to the Hall–Petch formula (Equation (2)) [18], the yield point is inversely proportional to the square root of the mean grain size, which confirms the previously formulated thesis.
f_y_ ~ 1/(d)^0.5^(2)

As confirmed during our own lab-tests, the value of the yield point remains practically unchanged in the temperature range up to approximately 600 °C. In higher temperatures, grains start to grow, which reduces the density of mesh of grain boundaries, and leads to a reduced yield of steel. In order to confirm a chemical composition of steel (Table 2), an EDS analysis was performed using a scanning electron microscope. The analysis showed the presence of the following alloy elements: carbon, iron, manganese, and chromium. The content of manganese (1.0 wt.%) and chromium (0.9 wt.%), Table 4, turned out to be slightly higher than the content specified in the metallurgical certificate [16] (Table 2).

#### 3.1.1. Microstructural Analysis of Samples Heated at 600 °C

After tempering at a temperature of 600 °C with both air-cooling and water-cooling methods, the martensitic microstructure of tempered steel is still preserved. Quantitative analysis of microstructure revealed no changes in size of martensite needles. This temperature, although it exceeds the nominal tempering temperature used during the production process of this grade of bolts, turned out to still be too low to initiate the phase transformation, consisting in the decomposition of martensite and the release of austenite, Table 5. Such an excess in comparison to the nominal tempering temperature is probably the result of the presence of alloying elements that may have caused certain disturbances in technological parameters in the case of repeated thermal treatment.

The analysis of the photos presented in Table 5 does not reveal any significant differences in the microstructure, neither along the width of the shank nor between the samples with different heating times or cooling methods.

Although there are no visual differences in the microstructure between the IS AC/600 and WC/600 samples, in the case of both the air-cooled and water-cooled bolts, the value of residual tensile strength after heating for 60 min turned out to be lower than the initial value of the IS bolts by approximately 12% and in the case of bolts heated for 240 min it was by as much as 22%. These tests showed the effect of the heating time on the value of residual strength properties of fire-exposed bolts.

#### 3.1.2. Microstructural Analysis of Samples Heated at 800 °C

Tempering at the temperature of 800 °C with air cooling (AC/800/60 and AC/800/240 samples) leads to the martensite 🡪 austenite phase transformation. Then, during the cooling phase, ferrite is released (light areas) at the boundaries of austenite grains (Table 6). The temperature of 800 °C is above the A_1_ line on the Fe-C phase diagram, therefore martensite austenitizes at this temperature.

The austenite transformation during the tempering of alloy steels is influenced by the content of elements dissolved during austenitisation. Chromium, which is an alloying element of the tested steel, noticeably increases its transformation temperature and as a result the amount of ferrite released during the heating at 800 °C is small, but it increases along with growing temperature and heating time. When analysing the photos of the microstructure shown in Table 6, it can be easily observed that the microstructure of the steel heated at 800 °C for 240 min is quite different from that heated for 60 min only. The grain size increases, the mesh of boundaries between respective phases loosens, which results in a further reduction in the residual tensile strength and hardness of the material and thus becomes more plastic. Chromium, present in the tested steel as one of the alloying elements in the range up to 1%, slightly reduces the hardness of ferrite and considerably increases its impact strength along with a simultaneous decrease in hardness as compared to the reference value at room temperature (HV = 324). Moreover, it increases the amount of plastic residual austenite in the quenching process. Since the concentration of carbon in ferrite is lower than in the austenite from which is released, the carbon content in austenite increases along with heating. After reaching the critical value, i.e., after exceeding the limit level of the solubility of carbon in austenite, the pearlite transformation begins and leads to the transformation of the remaining austenite into pearlite (dark areas), which results in obtaining a plastic and soft ferrite–pearlite structure.

The transformation into ferrite–pearlite structure is demonstrated by a reduction in the residual ultimate tensile strength (UTS) from the initial state value equal to 1001 MPa (maximum tensile force F_m_ = 245 kN) to the level of UTS = 595 MPa (maximum tensile force F_m_ = 146 kN), with HV = 200 for the heating time of 60 min and UTS = 585 MPa (maximum tensile force F_m_ = 143 kN), and with HV = 196 for the heating time of 240 min. It is worth noting here that in the case of samples heated at the temperature of 800 °C and air-cooled, despite the noticeably different microstructure (Table 6), the heating time does not have a significant effects on the differences in the value of residual tensile strength—in both cases it remains on an almost identical level. In the case of a longer heating time, recrystallization of the defective ferritic matrix was observed, which translated into a slightly greater decrease in the UTS value.

Tempering at the temperature of 800 °C combined with subsequent water-cooling (WC/800/60 and WC/800/240 samples) did not result in the martensite 🡪 ferrite + pearlite phase transformation, despite reaching the heating temperature above A_1_. As can be seen in the photos of the microstructure presented in Table 7, the material in this case still shows a typical martensite structure similar to the original one in the IS state. The cooling method applied, characterised by a high speed of thermal energy reception, had a significant impact on inhibition of the phase transformation.

Rapid water cooling of the steel heated up to austenitisation temperature made it harden again. The degree of hardening turned out to be clearly dependent on the heating time. The martensite transformation did not fully take place within 60 min (residual austenite is still visible), which resulted in the value of UTS = 1064 MPa (maximum tensile force F_m_ = 261 kN—higher than the reference value for the sample in the initial state), with a simultaneous significant increase in hardness to HV = 361. Heating for 240 min resulted in austenitisation of the whole microstructure and the martensite transformation in the entire volume of the material, which translated into the value of UTS = 1064 MPa (maximum tensile force F_m_ = 261 kN—analogous as in the case of the heating time of 60 min), with another significant increase in hardness to the level of HV = 543.

This stage of research showed that, in the case of shock water-cooled samples, the duration of the heating time had a significant effect on increased hardness of the material and the use of rapid cooling noticeably influenced the value of residual tensile strength, which in this case exceeded the reference value obtained for bolts in the initial state IS (UTS = 1001 MPa).

#### 3.1.3. Microstructural Analysis of Samples Heated at 1000 °C

Tempering at 1000 °C with air cooling (AC/1000/60 and AC/1000/240 samples, Table 8), similarly to the heat treatment at 800 °C, caused the martensite 🡪 austenite phase transformation during the heating process and then the release of ferrite and pearlite within the boundaries of austenite grains during the phase of slow and free cooling.

The share of ferrite (light areas) decreased noticeably, whereas the share of pearlite (dark areas) increased compared to samples heated at 800 °C and air-cooled (AC/800/60 and AC/800/240). A significant grain growth and recrystallization of the defective ferritic matrix were observed, which in turn resulted in a decrease in the residual ultimate tensile strength to the level of UTS = 619 MPa (maximum tensile force F_m_ = 152 kN), with HV = 204 for the heating time of 60 min and UTS = 576 MPa (maximum tensile force F_m_ = 142 kN) with a noticeably lower HV = 153 for the heating time of 240 min, respectively. Attention should be paid to a visible difference in hardness depending on the applied bolt heating time. In the case of samples cooling down naturally, an increased heating time results in a reduced value of residual tensile strength and hardness of the material. The same trend was also observed in the case of the AC/600 and AC/800 series samples, which may confirm that this trend is of a structured nature.

Heat treatment at the temperature of 1000 °C with water-cooling (WC/1000/60 and WC/1000/240 samples), just as in the case of heat treatment at 800 °C, did not lead to the martensite 🡪 ferrite + pearlite phase transformation, although a temperature considerably higher than A_1_ was reached. The high cooling rate made the steel harden again such that the material still had the martensite structure similar to the initial one IS (Table 9).

The high temperature of heat treatment contributed to the formation of coarse-grained martensite. The degree of hardening turned out to be significantly dependent on the heating time. Within 60 min, complete austenitisation took place and coarse-grained martensite was obtained in the entire volume, which resulted in a significant increase in the residual ultimate tensile strength value to the level of UTS = 1178 MPa (maximum tensile force F_m_ = 289 kN) and exceeded the reference value for the sample in the initial state with HV = 540. Heating for 240 min turned out to be too long and caused the grains to grow, which translated into a decrease in the UTS value to the level of 869 MPa (maximum tensile force F_m_ = 213 kN) with a simultaneous considerable decrease in hardness to HV = 210.

### 3.2. Testing with the Use of X-ray Diffraction (XRD)

Steel of type 32CrB3 contains 0.74% of chromium in its chemical composition. Chromium increases the temperature of the austenite transformation, slightly decreases ferrite hardness, increases the amount of residual austenite in the quenching process, and increases impact strength [19]. Heat treatment of steel, in which chromium is the alloying element, may contribute to the formation of carbides, which in the end has a noticeable effect on mechanical properties of steel. Carbides formed in steel are a hard and brittle phase. They are formed as a result of the solubility level of carbon in austenite and ferrite changing along with the temperature change. The presence of carbides in steel most often increases its hardness, yield point, and tensile strength. The carbides are also responsible for the secondary hardness effect, i.e., an increase in steel hardness during tempering. At the same time, their presence can have an adverse effect on impact resistance, ductility, and fracture resistance of steel [19]. Increased hardness and residual tensile strength in the case of the majority of samples heated at the temperature of 800 °C and 1000 °C might lead to a presumption that the presence of carbides in the material structure could be responsible for some of these changes. In order to exclude the presence of carbides in the steel structure before and after the heating process at the temperature of 1000 °C, XRD tests were carried out. The tests were carried out on both samples cooled naturally in air (AC) and those shock-cooled in water (WC). An analysis of the XRD results showed that obtained diffractograms had not differed in the number of peaks, but only in their intensity, which confirmed the fact that there were no carbides present in the steel structure before and after the heating process for the case of air-cooled and water-cooled samples. Considering the size of this article, the detailed results of XRD tests are not presented in the wider range.

### 3.3. Quantitative Analysis of the Microstructure

In respect to each of the phases shown in the pictures of the bolt microstructure, which were taken with the use of a digital light microscope (at the magnification of ×100) on the samples specified in Table 1, a quantitative analysis was carried out in order to determine the number of grains visible in the photo, their density, mean grain area, mean, and equivalent grain diameter. The analysis was performed separately for the ferrite and pearlite phases. The tests were aimed at confirming the previously observed qualitative change in the grain size in quantitative terms and aimed at linking these changes with changes in residual mechanical properties of the analysed samples. Due to limited funds, the analysis was performed only in respect to selected samples.

Collective results showing the dependence of microstructure indices, i.e., mean diameters of ferrite and pearlite grains, HV hardness, and tensile strength, on heat treatment parameters are summarised in Table 10 and Figure 1.

The measurements showed that in the case of air-cooled samples, the size of the pearlite grains increased along with an increase in the temperature level as well as the time of thermal exposure. However the grain growth was not uniform across the entire width of the bolt shank. For air-cooled samples, both in the case of ferrite and pearlite grains, the extension of the heating time at 800 °C from 60 min to 240 min resulted in an over threefold increase in the grain equivalent diameter. Prolonged annealing at the temperature of 1000 °C resulted in a further growth of pearlite grains to approximately 13.0 μm in diameter, with a simultaneous approximately 20% decrease in the ferrite grain size.

In the case of water-cooled samples, the ferrite grain size is clearly smaller than in the case of air-cooled samples corresponding to them in terms of heat treatment conditions. This confirms the previous observations made on the basis of the visual analysis of the pictures included in Table 9. The water shock cooling of the samples prevented the phase transformation of austenite into pearlite and inhibited the growth of ferrite grains. The impact of the shock cooling on the size of ferrite grains was noticeable—near the outer walls of the sample shank, the grain diameter is almost half the size of those near the bolt axis. In the case of air-cooled samples, along with an increase in temperature and heat-exposure time, a slight downward trend in the value of the residual tensile strength and the hardness of the bolt material was also clearly visible. The water shock cooling re-hardened the bolts and was followed by a sharp increase in both the UTS and HV values (Figure 1).

### 3.4. Analysis of the Correlation between Hardness and Residual Tensile Strength

The purpose of the analysis was to investigate veracity of the linear correlation described by the Formula (1) between tensile strength UTS and hardness HV of steel subjected to heat treatment during the production process and then subjected to secondary thermal treatment, e.g., as a result of exposure to thermal effects of a fire. The relationship (1) has been confirmed so far by numerous tests carried out almost exclusively on samples of commonly used structural steels working in normal thermal conditions. The available literature does not provide any information proving its correctness in relation to the value of residual tensile strength characteristic of the material of high-strength steel bolts after the fire exposure.

In order to better illustrate the effect of secondary heat treatment parameters on the relationship between hardness HV and residual tensile strength of the bolt steel, a comprehensive diagram has been presented in Figure 2. It shows a clear linear relationship between the values of hardness and tensile strength but it meets the criterion described by the scaling factor 3.2 ÷ 3.5 that is not in the entire domain of determinacy. In the case of air-cooled samples, the value of this factor obtained in the tests fluctuates in the range of 3.0 ÷ 3.8 and in the case of water-cooled samples the factor fluctuates in the range of 2.0 ÷ 4.1, respectively. In the case of air-cooled samples, attention should be paid to a noticeable trend of a decrease in strength (in relation to the reference value, characteristic of the material in its initial state) accompanying a temperature and heating time increase. The hardness parameter also shows a similar trend. In the case of shock water-cooled samples, this trend which is characteristic of air-cooled samples, is reversed. Along with an increase in the temperature and heating time, the values of the residual tensile strength and hardness of the bolt steels increase. Some regularity of this trend is locally disturbed only in the case of the WC/600/240 and WC/1000/240 samples, which may, however, be caused by a systematic error due to the small size of the sample. Confirmation of this thesis would nevertheless require further research. In the case of shock water-cooling of samples (simulation of a rescue and firefighting operation), both residual tensile strength and hardness of the bolt material increases significantly and under certain conditions even exceeds the reference values characteristic of the tested high-strength bolts in their initial state. On the one hand, it can be perceived as a desirable effect due to the significant strengthening of the material and, on the other hand, it can be perceived as a negative effect due to an increase in its brittleness, which translates into the possibility of an abrupt form of failure in the event of overloading the bolts in the joint during an operation phase after a fire.

In order to determine the effect of the cooling method, temperature level and heating time on the disturbance of the relationship between the hardness of the bolt material and its residual tensile strength and the relationships between these two values in different configurations are shown in Figure 3 and Figure 4.

When analysing the diagrams shown in Figure 3, it can be observed that a negative effect of long heating of a particular series of samples is clearly visible—both in the value of residual ultimate tensile strength UTS and hardness HV. The same trend showing the effect of the heating time on stability of the analysed material properties is visible for each temperature level. Excessively long heating results in a decrease in residual strength. As a consequence, material hardness is also reduced.

The analysis of Figure 4 shows that in the temperature range up to 600 °C, both in the case of air-cooled bolts and those water-cooled, no effect of the heating time on stability of the analysed mechanical properties of the samples has been identified. Corresponding pairs of graphs for the AC/600/60 and WC/600/60 and AC/600/240 and WC/600/240 samples do not differ from each other. The only difference between the bars of the graphs shown in Figure 3a,b is in the UTS and HV values. Even the UTS/HV inter-relationships for the corresponding pairs of samples are almost identical. This confirms the previous observations made on the basis of the analysis of microstructure pictures, described in Section 3.1.1 of this article.

An increase in the heating temperature to 800 °C and higher makes the differences between the air-cooled and water-cooled samples clearly visible. Under similar thermal conditions, water shock cooling results in achieving a much higher residual tensile strength of the sample material—UTS and hardness HV compared to the sample freely cooling in the air. This results from the fact that the samples are re-hardened and the processes of phase transformations are stopped.

## 4. Discussion

A comprehensive approach to the analysis of the effect of simulated natural fire conditions on microstructural changes in the material of construction bolts and also on their key strength properties, taking into account at the same time the effect of thermal conditions, exposure time, and cooling method, is presented in this paper and cannot be found in available literature. In particular, with regard to the analysis of the influence of exposure time on the given thermal conditions, the conducted research is somewhat unique. In this context, this work brings a completely new value to the state of knowledge in this area and may really contribute to the progress of work on methods of post-fire assessment of quality and reliability of structures.

The study confirmed that, as a result of fire exposure, the microstructural and mechanical characteristics of high-strength bolts undergo significant changes. These changes can have a negative impact on the safety and reliability of steel structures, as well as the manner they behave after a fire.

The bolts heated to a temperature exceeding 600 °C and cooled naturally in the air, as a result of fire exposure undergo a martensite 🡪 austenite phase transformation during the heating process and then ferrite and pearlite are released within the boundaries of austenite grains in their structure. This structural modification renders steel softer and bolt load-bearing capacity considerably lower. However, as a rule and at the expense of increased plasticity, an overloaded bolt does not rapidly fail. Bolts that underwent such a heating cycle permanently lose their original strength properties. This, in general, confirms the observations made by Kirby [1,2], Kodur et al. [5,6], and Yahyai [7]. However, each of them in their considerations ignored the effect of rapid cooling, which is a natural consequence of the fire-fighting action, as well as the related consequences concerning both the microstructure and mechanical properties of the tested bolts. Kodur et al. [5,6], similar to Yahyai et al. [7] did not focus on detailed microstructural studies, and they only tried to link the bolt failure model with the macroscopic image of the fracture surface. In the case of screws naturally cooled in the air after the annealing process at a temperature exceeding 600 °C, similar to observations of Sajid et al. [10], the significant growth of pearlite and ferrite grains was observed leading to a reduction in the ultimate tensile strength. This clearly confirms the truth of the Hall–Petch formula (Equation (2)) [18] not only in relation to the yield point of steel but also to the value of the UTS.

Bolts heated to a temperature exceeding 600 °C and shock cooled with water (e.g., during firefighting operation in a real fire) retain a martensite structure, similar to the original one, because they are re-hardened through rapid cooling. Due to the sudden reception of thermal energy, they do not undergo the microstructural change characteristic of the martensite 🡪 ferrite + pearlite phase transformation, despite reaching a temperature significantly higher than A_1_. The steel of the bolt hardens and the temporary load capacity of bolts can, as a result of the re-hardening, even exceed the load capacity that the bolt had in its initial state or be very close to this value. This is performed at the expense of increased brittleness of the bolt such that, in the event of overload, the bolt can be expected to fail rapidly. Although the conducted research did not carry out such detailed quantitative analysis as performed by Chi and Peng [9], the obtained results confirmed the essence of their observations with regard to the batch of water-cooled bolts.

The conducted research confirmed the possibility of using microstructural tests for a post-fire assessment of steel structures and an attempt to reconstruct fire scenarios. However, referring to the concept of the use of metallurgical changes occurring in the screw material for the diagnosis of buildings damaged by fire proposed, inter alia, by Kirby [1,2], Chi and Peng [9], and Sajid et al. [10], it should be stated emphatically that unless the precise fire temperature is known, an assessment based solely on microstructural tests may not be sufficient or may be flawed with significant errors. This may be the case, for example, when the image of the bolt material microstructure indicates a martensite structure. In the case of high-strength bolts, this indication is typical for bolts in their initial state, bolts heated to a temperature lower than 600 °C, and bolts heated to a much higher temperature and rapidly cooled with water. In this case, a reliable assessment cannot be made without conducting additional destructive strength testing. This example shows that the post-fire assessment of bolts should be performed in an extremely reasonable and careful method.

The strength tests of the initial phase (which for the sake of clarity of the article content have not been included here) confirmed the earlier observations made, among others, by Kirby [1,2] and Kodur et al. [5,6] that the residual mechanical properties of bolts subjected to fire, due to their heat treatment history, differ from those determined for carbon structural steels given in the standards for designing structures [20,21]. The resistance reduction factors given in Annex D of [20] applicable during connection design were determined from the fire tests of bolts conducted for British Steel [1,2] by Kirby. Although the reduction factors deduced from the tensile and shear tests were different, in order to simplify the design guidance the conservative results were applied for both types of connections—those loaded in tension and those loaded in shear. Theoretically, applying these guidelines to bolts can lead to conservative estimates but one has to realize that these provisions were based on test results of air-cooled bolts only. It cannot be forgotten that the increased brittleness of bolts and elements arising from the fire-fighting action increases the risk of an abrupt and progressive collapse caused by the inability of the structure to redistribute internal forces by creating local yielding areas. This, in turn, may result in an increase in the threat to lives and the health of users of the building, who are the potential victims of a fire and members of rescue teams. In the case of new construction design, the use of Annex D [20] in connection with design likely leads to safer estimates. The observation of existing objects destroyed by a fire but designed on the basis of earlier standards leads to completely opposite conclusions than discussed above. It is the connections in older buildings that seem to be the weakest link in the entire structure. The authors draw attention to the importance of this problem, especially in the case of facilities such as shopping centres, galleries, arcades, or other public facilities made of steel structures. However, it should also be noted that the quality of steel products has improved since Kirby’s publication of the results [1,2] and the reduction factors may need to be updated based on the results of up-to-date tests.

## 5. Limitations, Recommendations, and Suggestions for Further Research

The outcomes show that the method of carrying out a rescue and firefighting operation may be of key importance for the safety of structure and people staying in a building in fire. In the case of structures subject to dynamic loads, water cooling of bolts should be avoided as this will render them more susceptible to brittle fracture. In the case of fire-damaged structures that have already undergone significant deformations, potential gains and losses should be assessed on a somewhat real-time basis. If members have not been deformed by a fire and there is a real chance of renovating and re-using, then the risk of cooling the bolts may be taken in order to increase their load capacity in real time. Of course, during the reconstruction or renovation of the structure, such bolts must be replaced with new ones with predictable strength properties. If the structure cannot be saved, its stability is compromised or it has undergone significant deformations preventing it from being reused, bolts should not be hardened to permit their slow plastic failure at the moment of being overloaded.

A big challenge in the logistic, economic, and scientific sense is the study of complete joints and connections performed in their natural scale. The studies of butt joints [22,23,24] and lap friction joints [25] carried out so far and described in the literature focus mostly on establishing the model of failure and analysis of the behaviour of these joints under load. They also form the basis for the validation and verification of numerical models that try to map the physical behaviour of a connection or joint in a real structure. These studies do not take into account the drop in pre-stressing force as a result of fire in connections that were pre-stressed at the assembly stage. This gap is worth paying more attention to in terms of conducting completely new research. It is also worth considering, in the future, a wide range of tests utilizing stainless steel and fire-resistant bolts. The heat-resistant and creep-resistant steels are able to withstand temperatures up to approximately 1150 °C for a long period of time. They obtain their heat resistance thanks to a wide range of alloy additives: aluminium, chromium, silicon, molybdenum, vanadium, tungsten, titanium, and cobalt and increases the energy of interatomic bonds. In particular, unlike conventional carbon steels, austenitic alloys used for creep-resistant steels are characterised by extremely low susceptibility to structural changes in long-term operation at high temperatures. The differences in resistance to high temperatures between carbon steels and fire-resistant steels, in the context of their susceptibility to microstructural changes, are best illustrated by comparing the diagrams of their specific heat as a function of temperature. The characteristic peak at 735 °C in the case of carbon steels, resulting from their metallurgical changes, does not occur in the diagram for fire-resistant steels. Although the use of heat-resistant and creep-resistant steels in modern construction is currently rather niche, in just a few years, due to technical progress, the development of engineering of structural materials or technological methods for modifying the microstructure of construction materials [26,27] could become quite common.

## 6. Conclusions

The presented research is of a practical nature with great application potential for engineering practice.

The results obtained can be helpful for the purpose of assessing structures that have survived a fire without any major damage, in the context of the possibility of reusing selected elements, and those that have been damaged by fire. In the latter case, the results can be used to recreate a fire development scenario and to estimate maximum values of fire temperatures in a situation where they have not been measured by firefighting and rescue services.

The microstructural tests confirmed that under environmental conditions corresponding to a simulated fire situations and an accompanying firefighting operation, significant structural changes occur in the material of bolts that are strongly dependent on the temperature reached, the time of exposure to fire conditions, and the method of cooling. These changes result in modifications of residual strength properties and are crucial from the point of view of structural safety, in particular tensile strength and correlated hardness. Microstructural changes significantly affect how the bolt material behaves, which usually determines how the structure will fail in the event of a potential collapse.

The undertaken research and obtained results indicate that this work should be continued with focus on development of detailed guidelines for designing bolted joints while simultaneously taking into account the effects of fire.

## Figures and Tables

**Figure 1 materials-14-03116-f001:**
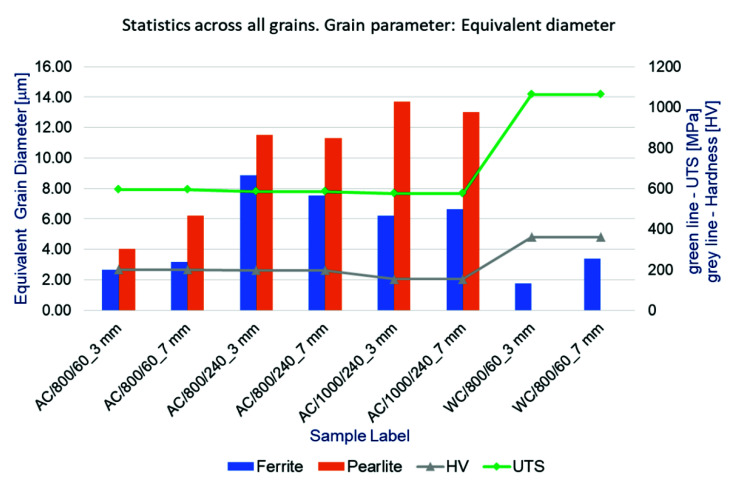
Diagram presenting dependence of microstructural indices, i.e., mean equivalent diameters of ferrite and pearlite grains, HV hardness, and ultimate tensile strength (UTS) on heat treatment parameters.

**Figure 2 materials-14-03116-f002:**
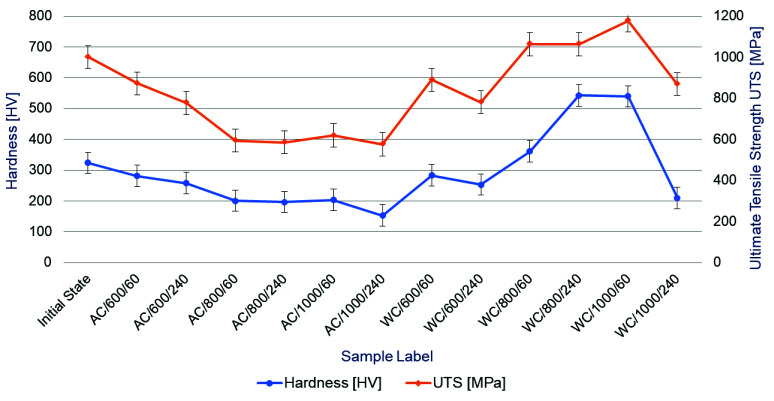
Diagram of correlations between hardness HV and residual tensile strength of the bolt steel after secondary heat treatment depending on parameters of this treatment.

**Figure 3 materials-14-03116-f003:**
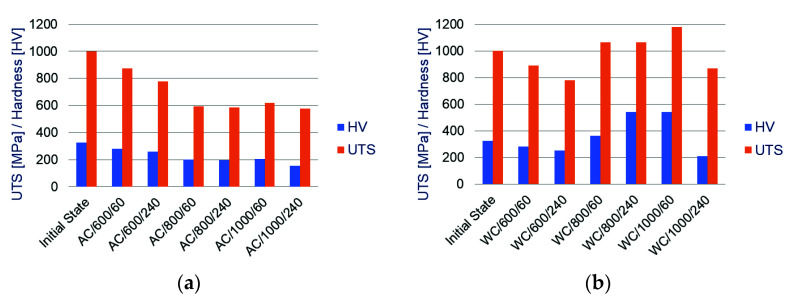
Diagram of dependencies between bolt steel hardness and bolt residual tensile strength on heating time for samples: (**a**) naturally air-cooled; (**b**) shock water-cooled.

**Figure 4 materials-14-03116-f004:**
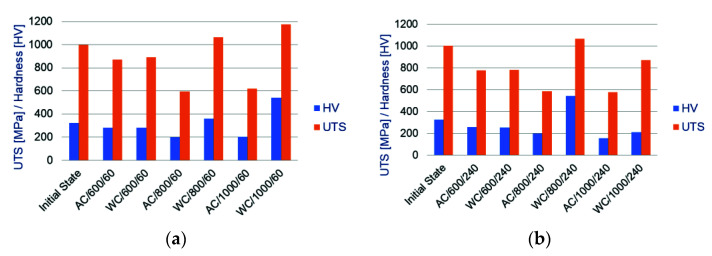
Diagram of dependencies between bolt steel hardness and bolt residual tensile strength on heating time for samples subjected to secondary heat treatment for the time of: (**a**) 60 min; (**b**) 240 min.

**Table 1 materials-14-03116-t001:** List of tested samples.

Labels of Samples Prepared for Testing	List of Samples Tested with the Use of Light Microscopy	List of Samples Tested for the Purpose of the Quantitative Analysis
IS—reference sample	Yes	No
AC/400/60	No	No
AC/400/240	No	No
AC/600/60	Yes	No
AC/600/240	Yes	No
AC/800/60	Yes	Yes
AC/800/240	Yes	Yes
AC/1000/60	Yes	No
AC/1000/240	Yes	Yes
WC/400/60	No	No
WC/400/240	No	No
WC/600/60	Yes	No
WC/600/240	Yes	No
WC/800/60	Yes	Yes
WC/800/240	Yes	No
WC/1000/60	Yes	No
WC/1000/240	Yes	No

**Table 2 materials-14-03116-t002:** Chemical composition of the 32CrB3 bolt steel according to the manufacturer’s quality certificate [16].

Steel Designation	Chemical Composition [%]										
32CrB3	C	Mn	Si	P	S	Cr	Ni	Cu	Al	Mo	Sn
	0.31	0.84	0.13	0.012	0.013	0.74	0.08	0.15	0.025	0.018	0.010

**Table 3 materials-14-03116-t003:** Microstructure of steel in its initial state.

Sample Label	Image at a Distance of 3 mm from the Sample Edge	Image at a Distance of 7 mm from the Sample Edge
IS—initial state (reference sample)	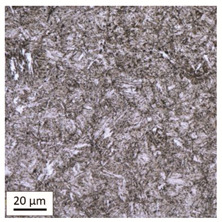	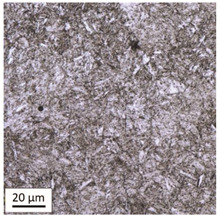

**Table 4 materials-14-03116-t004:** Microstructure of steel in its initial state along with the results of the EDS analysis.

Microstructure of Steel in the Initial State	Results of the EDS Analysis
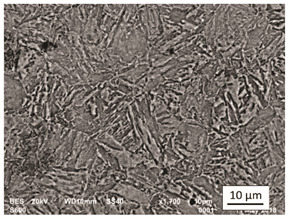	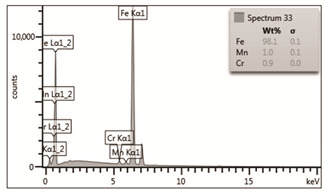

**Table 5 materials-14-03116-t005:** Microstructure of steel after heating at 600 °C for 60 min and 240 min with air-cooling and water-cooling.

Sample Label	Image at a Distance of 3 mm from the Sample Edge	Image at a Distance of 7 mm from the Sample Edge
AC/600/60	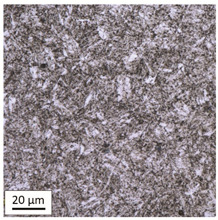	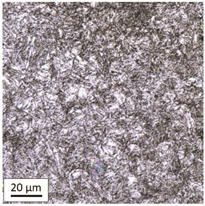
AC/600/240	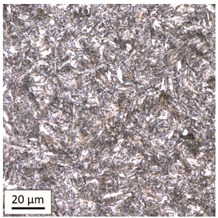	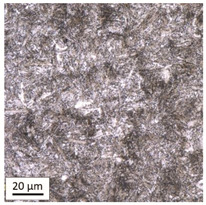
WC/600/60	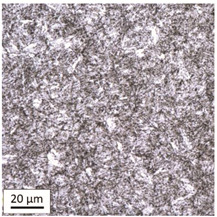	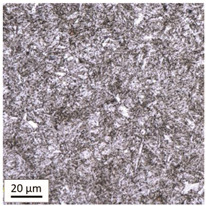
WC/600/240	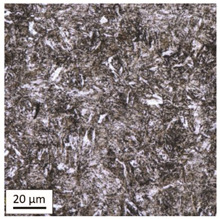	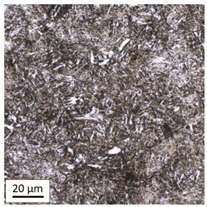

**Table 6 materials-14-03116-t006:** Microstructure of steel after heating at 800 °C for 60 min and 240 min with air-cooling.

Sample Label	Image at a Distance of 3 mm from the Sample Edge	Image at a Distance of 7 mm from the Sample Edge
AC/800/60	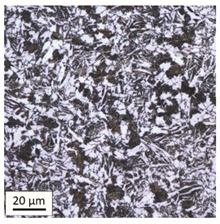	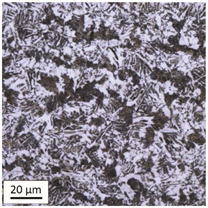
AC/800/240	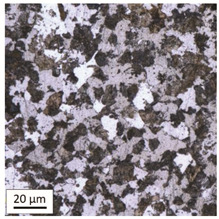	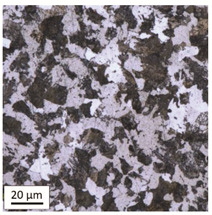

**Table 7 materials-14-03116-t007:** Microstructure of steel after heating at 800 °C for 60 min and 240 min with water cooling.

Sample Label	Image at a Distance of 3 mm from the Sample Edge	Image at a Distance of 7 mm from the Sample Edge
WC/800/60	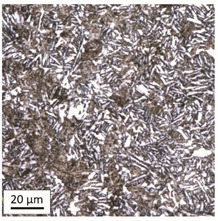	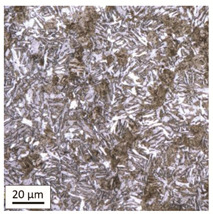
WC/800/240	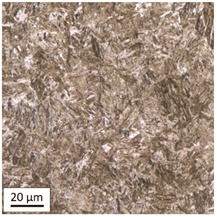	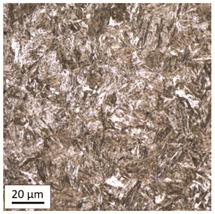

**Table 8 materials-14-03116-t008:** Microstructure of steel after heating at 1000 °C for 60 min and 240 min with air-cooling.

Sample Label	Image at a Distance of 3 mm from the Sample Edge	Image at a Distance of 7 mm from the Sample Edge
AC/1000/60	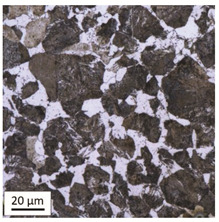	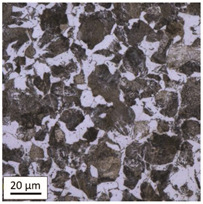
AC/1000/240	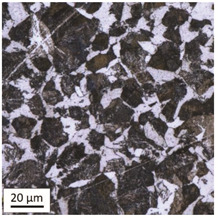	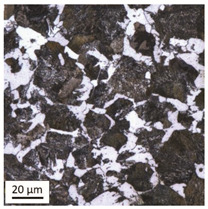

**Table 9 materials-14-03116-t009:** Microstructure of steel after heating at 1000 °C for 60 min and 240 min with water-cooling.

Sample Label	Image at a Distance of 3 mm from the Sample Edge	Image at a Distance of 7 mm from the Sample Edge
WC/1000/60	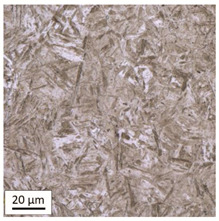	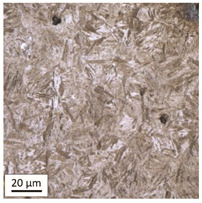
WC/1000/240	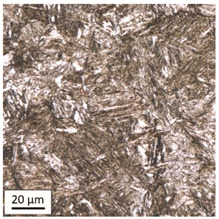	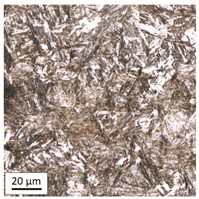

**Table 10 materials-14-03116-t010:** Results of measurements of the mean equivalent diameter of ferrite and pearlite grains, hardness, and strength of respective series of samples.

Sample Label	Equivalent Diameter [μm]		Hardness	UTS
	Ferrite	Pearlite	[HV]	[MPa]
AC/800/60_3 mm	2.67	4.01	200.41	594.53
AC/800/60_7 mm	3.19	6.20	200.41	594.53
AC/800/240_3 mm	8.85	11.50	196.46	585.47
AC/800/240_7 mm	7.55	11.30	196.46	585.47
AC/1000/240_3 mm	6.19	13.70	152.98	575.88
AC/1000/240_7 mm	6.62	13.00	152.98	575.88
WC/800/60_3 mm	1.77	not measured	361.26	1063.90
WC/800/60_7 mm	3.39	not measured	361.26	1063.90

## Data Availability

Data sharing not applicable.

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
