# Peer review of "Effect of Fire Temperature and Exposure Time on High-Strength Steel Bolts Microstructure and Residual Mechanical Properties"

_materials, 2021, doi:10.3390/ma14113116_

Round 1
Reviewer 1 Report
Comparative analysis showed a high effect of factors as the temperature level of the simulated fire, its duration and the fire-fighting method on the mechanical properties of bolts. It will be good if you will advise Fire resistant bolt as we have Fire resistant steels?
Reviewer 2 Report
The main objective was to assess the influence temperature, heating time and cooling method on microstructure and residual strength properties of tempered steel. The paper is well organized and structured. However, there are some aspects that need to be addressed. The following comments are raised to improve the quality of the present paper.
-The Introduction provide the necessary background information, but the number of references could be improved. Authors only present 17 references, which represents a good number but could be improved and with more recent works.
-Tables 9 and 10 should not be appear divided.
- The presented graphics may have better quality (font and size type).
-Also, the graphics presentation always must have the same dimension on the page, or only one column or always in two columns (side by side).
- What is the opinion of the authors about the influence of the steel specific heat peak on the obtained results for the samples at high temperatures?
-What is the possible evaluation to be made for different grades of bolt categories?
-If possible, authors should divide the chapter ‘’conclusions and recommendations’’. The last chapter will be ‘’Conclusion’’, where the future directions and research gaps need to be included. In previous could appear a chapter with ‘’Limitations and recommendations’’.
-The Introduction provide the necessary background information, but the number of references could be improved. Authors only present 17 references, which represents a good number but could be improved and with more recent works.
-If possible, authors should divide the chapter ‘’conclusions and recommendations’’. The last chapter will be ‘’Conclusion’’, where the future directions and research gaps need to be included. In previous could appear a chapter with ‘’Limitations and recommendations’’.
Reviewer 3 Report
- The current study investigates the influence of fire temperature and exposure time on the mechanical performance and microstructural changes in high steel tempered bolts used in public facilities. The authors do not clearly state what was done and what were the main findings.
- The abstract is poorly written, it is difficult to understand what was done in this study and what were the main findings.
- Please consider reviewing the abstract and highlight the novelty, major findings and conclusions.
- Line 17 “after they were properly prepared” this is a vague sentence and does not clearly tell us what is this proper preparation!
- Line 21 “As a result of the tests” this is also not necessary wording, please check the English level of the manuscript as the abstract does not read well so far.
- Line 26 “significant effect of all such factors as the temperature level of the simulated fire” again very generic sentence and does not tell us what were the findings!
- After line 128 please answer the following question: What is the research gap did you find from the previous researchers in your field? Mention it properly. It will improve the strength of the article.
- Table 2 needs a reference
- Please combine smaller paragraphs into larger ones such as in lines 173-181
- Please remove Table 3 and Table 4, you can use Table two and add more columns and put a tick mark or yes/no to indicate which ones were inspected under optical microscopy and quantitative analysis
- Line 220 the authors did not even mention they inspected hardness in the abstract..
- Lines 233-240 combine into larger paragraphs
- Kline 268 “As was also confirmed by authors’ own observations” these are very long and not meaningful sentences
- Extensive editing of English language and style required
- Combine lines 268-277
- Line 255 “the critical temperatures may somewhat vary” again very vague and generic sentences, until now I don’t see any critical discussion about the observed results
- Line 283 “does not noticeably change the microstructure of steel” I don’t understand what do author mean by noticeable.
- I think this paper lacks any critical discussion or analysis, the language used in explaining the results lacks any strong scientific content. All claims made are vague and do not imply a trend or an interesting observation
- Line 406 please support this fact with a reference unless you found that on your own
- Line 413 same as above
- Line 454 “the grain size slightly decreased” I really don’t understand the way the authors describe their findings, slightly is not a scientific word to describe a result…
- Please add error bars on Figure 2 results
- Figure 3 is missing units
Round 2
Reviewer 3 Report
The authors have answered all questions and manuscript quality have improved paper can be accepted
but please consider the following
combine all the tables which says Labels of tested samples in one table, it is really waste of table space to add them seprately in different tables.
Author Response
Dear Sir or Madam,
Let us thank you again for your insightful review of the article, improving its quality.
Please kindly note that this has been done according to your earlier comment contained in the review of Round 1 (please see below). Former Table 3 and Table 4 have been merged into Table 1 and deleted. This may have gone unnoticed as *.doc and *.pdf files of the manuscript available on the Publisher's website, for reasons we do not understand, differ from each other and cause unnecessary confusion. In the available *.pdf version, there are still separate tables showing the state before the change. Please kindly check the *.doc version of the manuscript. We do not know if the system will enable such an option, but for you information we will try to attach the current version of the *.pdf file, consistent with the editable version, as the attachment.
Please remove Table 3 and Table 4, you can use Table two and add more columns and put a tick mark or yes/no to indicate which ones were inspected under optical microscopy and quantitative analysis
Answer/Comment of the Authors:
Has been done, according to your advice.
